

# Two $T$-linear scattering-rate regimes in the triangular lattice Hubbard model

Jérôme Fournier[1], Pierre-Olivier Downey[1], Charles-David Hébert[1], Maxime Charlebois[1,2] and A.-M. S. Tremblay[1]

**1** Département de physique, RQMP and Institut quantique, Université de Sherbrooke, Québec, Canada J1K 2R1
**2** Département de Chimie, Biochimie et Physique, Institut de Recherche sur l'Hydrogène, Université du Québec à Trois-Rivières, Trois-Rivières, Québec G9A 5H7, Canada

## Abstract

In recent years, the $T$-linear scattering rate found at low temperatures, defining the strange metal phase of cuprates, has been a subject of interest. Since a wide range of materials have a scattering rate that obeys the equation $\hbar/\tau \approx k_B T$, the idea of a universal Planckian limit on the scattering rate has been proposed. However, there is no consensus on proposed theories yet. In this work, we present our results for the $T$-linear scattering rate in the triangular lattice Hubbard model obtained using the dynamical cluster approximation. We find two regions with $T$-linear scattering rate in the $T$—$p$ phase diagram: one emerges from the pseudogap to correlated Fermi liquid phase transition at low doping, whereas the other is solely caused by large interaction strength at large doping.



# 1  Introduction

At the lowest temperatures in any metal, when the phonon contribution becomes negligible, one expects a Fermi liquid with $T^2$ resitivity. Although it is indeed the case in most materials, many do not abide by this rule, having instead a linear in temperature scattering rate. This is the case for a wide variety of materials, such as twisted bilayer graphene [1–3], transition metal dichalcogenides [4], pnictides superconductors [5], heavy fermions [6–8], organic superconductors [5] and cuprates [9–11]. This kind of behavior is even found theoretically in the square lattice Hubbard model [12] and in the Sachdev–Ye–Kitaev model [13].

$T$-linear scattering rate is often the result of electron-phonon scattering. This is the case for example in copper and twisted bilayer graphene [14]. However, at temperatures lower than the Debye temperature, this mechanism can no longer explain $T$-linear scattering. $T$-linear scattering rate must then be caused by another type of mechanism.

Metals that exhibit $T$-linear scattering rate at high temperature, beyond the Mott-Ioffe-Regel limit $k_F \ell \sim 1$, are called bad metals [15–17]. When the linear regime extends asymptotically close to $T = 0$, we refer to strange metal behavior. Cuprates are a nice case study of strange metals since their scattering rate has been thoroughly studied from the day of their discovery [18,19]. In addition, their $T$-linear scattering rate spans a large portion of the cuprate's phase diagram, sometimes up to high temperatures [18,19].

The idea of a universal limit on the scattering rate was presented to explain the $T$-linear scattering rate [20]. Using Drude's formula to find the relaxation time $\tau$, it has been observed that many strange metals obey the simple equation $\frac{\hbar}{\tau} = \alpha k_B T$, where $\alpha$ is between 0.7 and 1.1 [21–23]. The idea that this universal law could also be applied to very different materials with very similar values of $\alpha$ has led some to believe that electrons are subject to a universal Planckian limit of $\alpha \sim 1$ [10, 24–29].

The close proximity of strange-metal behavior to optimal doping in cuprates has led some to believe that understanding it could be the key to uncovering the mechanism behind superconductivity in hole-doped cuprates [30, 31]. The $T$-linear dependence of the scattering rate in cuprates is still a subject of research [32–35].

In this work, we present the phase diagram and the temperature-dependent scattering rate on the hole-doped triangular-lattice Hubbard model using the dynamical cluster approximation (DCA) [36] for the six-site cluster shown on Fig. 1a). DCA is a cluster extension of dynamical mean-field theory (DMFT) that is particularly suited for doped Mott insulators in regimes where long-wavelength particle-particle and particle-hole fluctuations are negligible.

The geometrical frustration inherent to the triangular lattice is particularly useful to suppress the above-mentioned fluctuations, making the thermodynamic limit reachable at finite temperature on small lattices. Our three main results are as follows:

First, our most unexpected finding is the observation of $T$-linear electron scattering in two distinct regions of the phase diagram: one at low dopings and another at higher dopings. We attribute the former to doped-Mott insulator physics, showing that $T$-linear scattering at low doping is linked to the metal-to-pseudogap first-order transition known as the *Sordi transition* [37–39]. We refer to this regime as the Mott-driven $T$-linear scattering rate. Conversely, at higher dopings, we propose that the $T$-linear scattering is solely governed by strong interactions, occurring very far from the Mott transition. We refer to this region as the interaction-driven $T$-linear scattering rate. It is noteworthy that in both cases, we noted in Table 1 summarizes which of the characteristics usually associated to strange metals is respected in each regime. In addition, it is important to point out that we compute the electron scattering rate, not the transport scattering rate that would necessitate vertex corrections.

Second, the role of long-wavelength magnetic fluctuations is not important in either regimes since, at the temperatures that we can reach, frustration on the triangular lattice limits their effect. Indeed, for values of $U$ that we explore, close to the Mott transition, it has been found that even at half-filling magnetic order is not apparent [40–46] until, perhaps, very low temperature [47].

Third, we find that even on the triangular lattice, the quasiparticle scattering rate of the interaction-driven $T$-linear scattering rate is very near the Planckian result ($\alpha \sim 1$). We do not claim that Planckian scattering is a fundamental limit.

Although our work may be related to the fundamental physics that drives the strange metal in cuprates, our model most likely represents what would be seen in doped $\kappa$-ET structured doped organic superconductors [48,49], field-effect doped organic superconductors [50], silicon triangular lattice simulators [51], 1T-TaS$_2$ [52] or cold atoms experiments [53–55]. Nevertheless, we find it valuable to draw comparisons with cuprates, given their extensive history of exploration.

In the following sections, we discuss the model, then uncover the phase diagrams that will drive our discussion of the two possible $T$-linear scattering rate regimes.

## 2 Methodology

Here we present the model, then discuss the method that we use, and finally, comment on observables of interest.

### 2.1 Model

We capture the complex interplay between kinetic energy and potential energy of electrons on a lattice with the one-band Hubbard model [56–59]. The Hamiltonian is given by

$$\mathcal{H} = -\sum_{i,j,\sigma} t_{ij} c_{i\sigma}^{\dagger} c_{j\sigma} + U \sum_i n_{i\uparrow} n_{i\downarrow} - \mu \sum_{i\sigma} n_{i\sigma}, \tag{1}$$

where $c_{i\sigma}^{\dagger}$ and $c_{i\sigma}$ are respectively the creation and annihilation operators on site $i$ with spin $\sigma$, $n_{i\sigma}$ is the number operator, $t_{ij}$ is the kinetic energy associated to a hopping between sites $i$ and $j$, $U$ is the on-site Coulomb repulsion, and $\mu$ is the chemical potential. We work in natural units, thus interatomic distance $a$, Planck's $\hbar$ and Boltzmann's $k_B$ constants are unity, as is $|t|$ the nearest-neighbor hopping and the lattice spacing.

The lattice is shown on Fig. 1a). We take $t = t' = -1$ so that the lattice is triangular and that the Fermi surface is centered at (0, 0). Another hopping $t'$ crossing the one illustrated

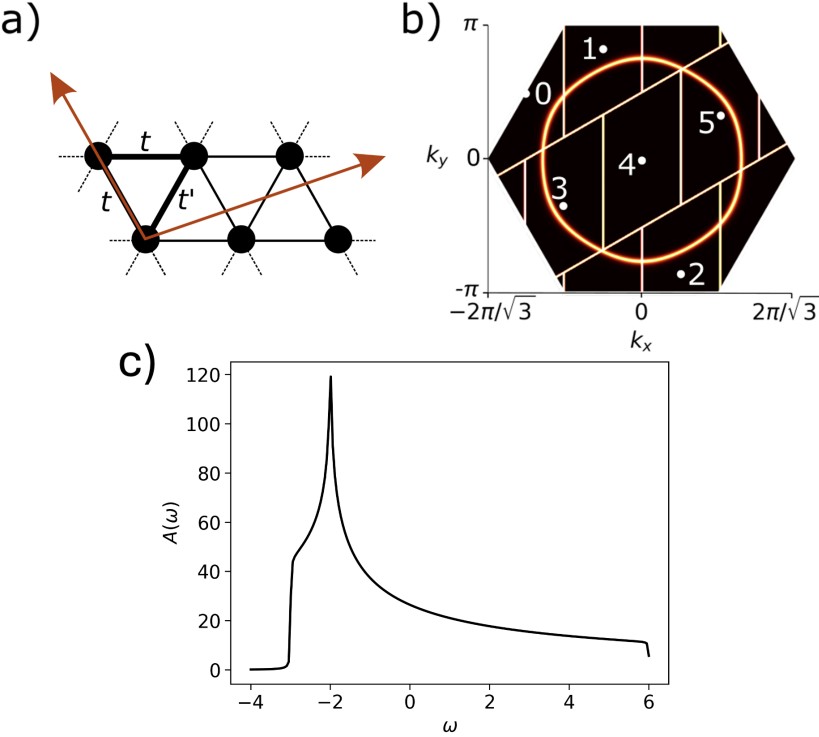

Figure 1: a) Hopping terms on the triangular lattice b) Fermi surface for $U = 0$ and $n = 1$ at $T = 0.1$ on the triangular lattice. The different patches used in the Brillouin zone of the triangular lattice and on the proxy square lattice made of the reciprocal lattice-vectors are illustrated. The superlattice vectors in red illustrate the periodic boundary conditions. Although $t' = t$ is satisfied in our work, this connectivity corresponds to a bipartite lattice when $t' = 0$. The illustrated Fermi surface is a hole Fermi surface. c) Local density of states for the non-interacting triangular-lattice.

would transform this problem into the problem of cuprates. We will later discuss implications of our results for cuprates. With these values of $t$ and $t'$, the non-interacting dispersion relation is:

$$\epsilon_{\mathbf{k}} = -2 \left[ t \cos(k_x) + t \cos\left( \frac{k_x}{2} - \frac{\sqrt{3}k_y}{2} \right) + t' \cos\left( \frac{k_x}{2} + \frac{\sqrt{3}k_y}{2} \right) \right].$$

The band parameters are the same for both hole-doping (denoted by $p$) and electron-doping (denoted by $x$) with respect to half-filling. Doping is controlled by the chemical potential. We focus mostly hole doping.

## 2.2 Solving the model

References [60, 61] have shown that in DCA, a six-site cluster impurity in a bath [36, 62–64] describes the same complex physics as the larger 12-site cluster at temperatures that are reachable near the Mott transitions. This is discussed further in Appendix A. For this reason, we use the same six-site cluster as in Ref. [60], defined by the superlattice vectors $R_x = (3, 1)$ and $R_y = (2, 0)$ as shown on Fig. 1a). Periodic boundary conditions in DCA impose that the Brillouin zone be separated into patches, one for every site on the impurity, their shape being just another degree of freedom [65, 66]. Fig. 1b) presents the layout we use. To illustrate how the Fermi surface is distributed among the patches we chose, the non-interacting Fermi surface is also displayed.

In DCA, one starts with a guess for the non-interacting cluster Green's function

$$\mathcal{G}_{0,\sigma}(i\omega_n, \mathbf{K}_i) = \frac{1}{i\omega_n - \bar{\epsilon}_{\mathbf{K}_i} + \mu - \Delta_\sigma(i\omega_n, \mathbf{K}_i)}, \tag{2}$$

where we define $\bar{\epsilon}_{\mathbf{K}_i} = \sum_{\tilde{\mathbf{k}}} \epsilon_{\mathbf{K}_i + \tilde{\mathbf{k}}}$, with $\sum_{\tilde{\mathbf{k}}}$ the sum on every $\tilde{\mathbf{k}}$ in a patch, and where we have, for $\mathbf{k}$ on the full Brillouin zone, $\epsilon_{\mathbf{K}_i + \tilde{\mathbf{k}}} = \epsilon_{\mathbf{k}}$ with $\epsilon_{\mathbf{k}}$ the bare band dispersion. The quantity $\omega_n$ denotes the $n^{\text{th}}$ fermionic Matsubara frequency, defined as $\omega_n = \frac{(2n+1)\pi}{\beta}$, with $\beta$ the inverse of temperature. Finally, $\Delta_\sigma(i\omega_n, \mathbf{K}_i)$ is the hybridization function, linking the bath and the impurities.

To find the cluster Green's function $\mathcal{G}_{c,\sigma}(i\omega_n, \mathbf{K}_i)$, one sends the non-interacting Green's function to an impurity solver. Here we use the continuous-time auxiliary-field (CT-AUX) [63, 67] quantum Monte-Carlo impurity solver because it scales well with the cluster size. Using the Dyson equation, one can extract the cluster self-energy

$$\Sigma_{c,\sigma}(i\omega_n, \mathbf{K}_i) = \mathcal{G}_{0,\sigma}^{-1}(i\omega_n, \mathbf{K}_i) - \mathcal{G}_{c,\sigma}^{-1}(i\omega_n, \mathbf{K}_i). \tag{3}$$

Projecting the lattice Green's function on the patches

$$\mathcal{G}_{loc,\sigma}(i\omega_n, \mathbf{K}_i) = \sum_j \frac{1}{i\omega_n - \epsilon_{\mathbf{K}_i + \tilde{\mathbf{k}}_j} + \mu - \Sigma_{c,\sigma}(i\omega_n, \mathbf{K}_i)} \tag{4}$$

($\tilde{\mathbf{k}}_j$ are the wave vectors inside the patch $\mathbf{K}_i$), leads to the self-consistency condition $\mathcal{G}_{loc,\sigma}(i\omega_n, \mathbf{K}_i) = \mathcal{G}_{c,\sigma}(i\omega_n, \mathbf{K}_i)$ from which the hybridization function necessary for the next iteration can be obtained:

$$\Delta_\sigma(i\omega_n, \mathbf{K}_i) = i\omega_n + \mu - \mathcal{G}_{loc,\sigma}^{-1}(i\omega_n, \mathbf{K}_i) - \Sigma_{c,\sigma}(i\omega_n, \mathbf{K}_i). \tag{5}$$

Substituting into the non-interacting Green's function Eq. 2, the next iteration of the DCA calculation begins.

Since the Green's function is symmetric in spin, we drop that index. We use the converged solution given by the data compilation algorithm proposed in Ref. [60].

Since DCA is a coarse-grained method, the momentum dependence of observables $\mathcal{O}$ are averaged over patches. This means that observables on a given patch $\mathbf{K}_i$ are obtained with the following equation

$$\mathcal{O}(\mathbf{K}_i) = \frac{1}{N} \sum_j \mathcal{O}(\tilde{\mathbf{k}}_j), \tag{6}$$

where $\tilde{\mathbf{k}}_j$ are the wave vectors inside the patch $\mathbf{K}_i$ and $N$ is the number of $\tilde{\mathbf{k}}_j$ inside the patch. Thus, the Green's function and the self-energy are constant within each patch $i$. Dividing out the Brillouin zone into six patches $\mathbf{K}_i$, the symmetries of the triangular lattice impose that $\mathcal{O}(\mathbf{K}_1) = \mathcal{O}(\mathbf{K}_2)$ and $\mathcal{O}(\mathbf{K}_3) = \mathcal{O}(\mathbf{K}_5)$. The patches are identified on Fig. 1b.

## 2.3 Observables

One of the important observables that we consider is the local scattering rate $\Gamma = 1/\tau$, where $\tau$ is the electron lifetime. This quantity is extracted from the local self-energy as

$$\Gamma = 1/\tau = -\text{Im}\left( \sum_i^{N_c} \Sigma(\omega = 0, \mathbf{K}_i) \right). \tag{7}$$

To obtain $\Sigma(\omega = 0, \mathbf{K}_i)$, we perform the analytical continuation using a simple polynomial fit on the first three Matsubara frequencies of $\text{Im}\Sigma(i\omega_n, \mathbf{K}_i)$, and extrapolate the polynomial to

Table 1: Table summarizing the similarities and differences between the usual strange metal, whose properties appear on the top row, and the two $T$-linear scattering rate regimes in this paper, namely Mott-driven and interaction-driven.

| Strange metal characteristics→ | $1/\tau \sim T$ | $1/\tau \sim T$ as $T \to 0$ | $1/\tau \sim T$ as $T \to \infty$ | $\omega/T$ scaling | Planckian dissipation | Extended range of doping | Isotropic scattering rate |
|---|---|---|---|---|---|---|---|
| **Mott-driven** ($p \approx 4\% \sim 6\%$) | ✓ | ✗ | ✓ | ✓ | ✗ | ✓ | ✗ |
| **Interaction-driven** ($p \approx 20\% \sim 30\%$) | ✓ | ✗ | ✗ | ✗ | ✓ | ✓ | ✓ |

$i\omega_n = 0$. In Appendix B, we show how the results are affected by the choice of polynomial order.

Here, we mostly focus on the electron scattering rate given by Eq. 7 instead of the quasiparticle scattering rate that would be obtained by multiplying Eq. 7 by the quasiparticle renormalization factor $Z$. This is because, even if the density of states presents a quasiparticle peak, some suggest that the quasiparticle picture breaks down in the strange metal [68]. We find that the exponent $n$ for the temperature dependence of the scattering rate does not change significantly when comparing electron scattering rate with quasiparticle scattering rate. There is however a sizable change in the slope caused by $Z$.

## 2.4 Limiting factors

The largest limiting factor of our method is the fermion sign problem that grows exponentially with the inverse temperature $\beta$ and the free energy of the system [69]. This limits the maximal size of the cluster as well as the value of $U$. On top of the fermionic sign problem, we are also limited in temperature by the low acceptance rate of Monte-Carlo configurations at low temperatures. Because of that, it is impossible to reach $T$ below 0.02 for the range of interactions we're interested in.

Another limiting factor at high temperature comes from the method used to perform the analytical continuation of the observable. Indeed, as the temperature is increased, the interval between Matsubara frequencies increases, leading to inaccurate polynomial fits. For this reason, we limit ourselves to $T$ lower than 0.2.

For a typical value of hopping $t$ in the cuprates of 0.3 eV, the range of temperature achievable with DCA would then be approximately between 70K and 700K. In BEDT organics the corresponding scales would be ten times smaller.

Finally, DCA is a coarse-grained method. Because the observables are averaged over patches, the momentum resolution is limited.

## 3 Results

We compute the scattering rate as a function of temperature for various hole-dopings of the Mott insulator. Doing this for many dopings, we build two temperature-doping phase diagrams[1] where we summarize the temperature dependence of $\Gamma = 1/\tau$ by color-coding the local exponent $n$ obtained form a local fit of the form $1/\tau = \alpha T^n + b$ on the data, as described in Appendix C.

---

[1] Note that Fig.2 does not strictly speaking display phase diagrams, but we use this name for convenience.

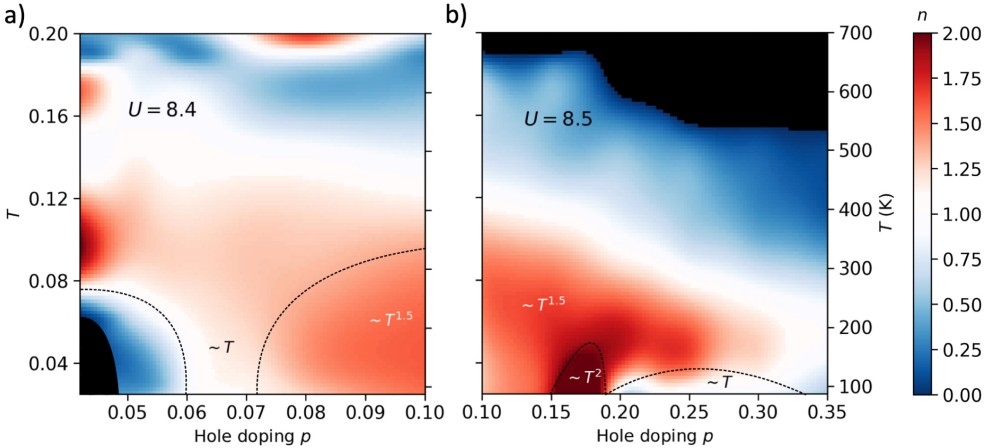

Figure 2: a) Temperature-doping phase diagram of the local scattering rate, defined by Eq. (7), for $U = 8.4$. Color coding represents the value of $n$ obtained, as described in Appendix C, from a local fit of the form $1/\tau = \alpha T^n + b$ of the scattering rate. No exponent was computed in the dark region near $p = 0.04$. The dashed line between $p = 0.04$ and $p = 0.06$ represents the temperature where the scattering rate starts to fall rapidly with temperature, whereas the one between $p = 0.07$ and $p = 0.1$ delimits the region where the scattering rate is proportional to $T^{1.5}$. b) Corresponding data for $U = 8.5$ in the high-doping range. The dashed line between $p = 0.15$ and $p = 0.18$ represents the temperature where we find a $T^2$ dependent scattering rate, whereas the one between $p = 0.18$ and $p = 0.34$ delimits the region where we find $T$-linear scattering rate at high dopings. This region of linearity between $p = 0.18$ and $p = 0.34$ appears very small on this figure because interpolation became difficult at lower temperature. Fig. 5 shows that the data extends to $T = 0.02$ and continues to exhibit linearity. Parts a) and b) share the same vertical axis. This means that the temperature range for both figures is the same. Note that all dotted lines are only guides to the eye. The temperature scale is fixed by taking $t = 0.3$eV, typical value for cuprates

We choose values of interaction $U$ slightly lager than the critical value of $U$ for the Mott transition at half-filling ($U \approx 8.2$ for $T = 0.15$ [60]). The first diagram on Fig. 2a) obtained at $U = 8.4$, focuses on the low doping behavior. The second, presented on Fig. 2b) for $U = 8.5$, focuses on high dopings. At low dopings, the value of $U$ is chosen slightly smaller because lowering $U$ increases the average sign in the Monte-Carlo calculations and makes it possible to converge in the pseudogap regime at slightly lower temperatures. The raw scattering rates that we used to draw these phase diagrams as a function of temperature at $U = 8.4$ and $U = 8.5$ are displayed respectively on Figs. 3 and 5. The raw data extends to slightly lower temperature than that presented in Figs. 2a) and b). Figures 2a) and b) present the average scattering rate over patches, namely the local scattering rate Eq. (7).

The results between 10% and 15% hole doping in Fig. 2b) exhibit a $T^{1.5}$ dependence of the scattering rate, qualitatively different from that found in organics [48,70] or in cuprates [9, 10, 22, 24]. Nevertheless, both doping regions illustrated in Figs. 2a) and 2b) display $T$-linear scattering rate for different ranges of temperature. Indeed, we find $1/\tau \sim T$ in Fig. 2a) for a wide range of temperature for $p$ near 0.06, while in Fig. 2b), we find $T$-linear scattering for hole dopings between 0.2 and 0.3, from $T \approx 0.03$ down to the lowest temperature achievable. This leads us to conclude that two different mechanisms are responsible for the $T$-linear scattering rates.

In the following sections, we present the two different regimes of $T$-linear scattering rate. In the first section 3.1, we show that the low-doping $T$-linear scattering rate is deeply rooted in the existence of the Sordi transition, the same pseudogap-metal first-order transition that is continuously connected to the Mott transition as reported in Ref. [38]. We thus use the name Mott-driven $T$-linear scattering rate, even though superexchange also plays a role in the Sordi transition, as can be argued from the fact that single-site DMFT finds a direct insulator to metal transition with doping [71]. Then, in section 3.2, we show that interactions seem to be the sole driver of the high doping $T$-linear scattering rate, thus the name interaction-driven $T$-linear scattering rate.

Both regimes of $T$-linear scattering rate found in this research share similarities with the strange metal phase found in cuprates. A list of these similarities can be found in Table 1. However, since both regimes have $T$-linear scattering rates that extrapolate to negative values at $T = 0$, we know that the $T$-linear scattering rate cannot be sustained at $T \to 0$. Because of this, we do not use the term strange metal to describe our findings. We instead use $T$-linear scattering rate.

## 3.1 Mott-driven $T$-linear scattering rate

Fig. 2a) displays the first region where we find $T$-linear scattering, what we call the Mott-driven $T$-linear scattering rate. This region spans a large area of the phase diagram, and goes down to the lowest temperatures near $p = 0.065$. A clearer picture emerges from Fig. 3, where we present the scattering rate as a function of temperature and doping for patches $\mathbf{K}_0$, $\mathbf{K}_1$ and $\mathbf{K}_3$.[2] At $p = 0.06$, we see in Fig. 2a) and Fig. 3 that $T$-linear scattering rate ranges from the lowest achievable temperatures to around $T = 0.2$, although we find two temperatures where there is a slight deviation from the $T$-linear regime, at seen in Fig. 2a) for $T \approx 0.08$ and $T \approx 0.18$. The raw data for the scattering rate at $p = 0.06$ in Fig. 3, shows that both deviations from the $T$-linearity are barely noticeable. At $T \approx 0.08$, the deviation is very similar to what is found in LSCO [10]. In the case of the higher temperature deviation, it is only barely noticeable in Fig. 3, indicating that this might be due to the fitting procedure (App. C) used for calculating $n$.

Other caracteristics of this regime include that the scattering rate in this $T$-linear scattering rate region is not isotropic, meaning that the scattering has a $\mathbf{K}_i$ dependence. Furthermore, the value of $\alpha$ for the quasiparticle scattering rate in this regime is larger than unity, making this regime non-Planckian, if we take a strict definition with $\alpha = 1$. In some experiments, Planckian dissipation is used as long as the value of $\alpha = 1$ is within experimental uncertainty, for example $\alpha = 1.2 \pm 0.4$ in Ref. [23]. In our case uncertainties are much smaller for a given choice of analytic continuation (See appendix B). Since our model does not include phonons, it is excluded that the $T$-linear scattering found at high temperature near $p = 0.06$ is a result of electron-phonon scattering at $T > T_D$.

Away from the optimal doping for the $T$-linear scattering rate, the behavior changes rapidly. For dopings lower than $p = 0.04$ in Fig. 3, we find an upturn in the scattering rate. This upturn is characteristic of the pseudogap phase. On the other hand, when $p$ increases to value larger than 0.07, we find, at low-$T$, a $T^{1.5}$ scattering rate. Since the low-doping $T$-linear scattering rate occurs at low temperature only for a very specific doping, it is likely to arise from a quantum-critical point. For $U = 8.4$, this quantum-critical point would be located near $p^* = 0.06$.

---

[2]We do not show the results for the patch 4 because it only accounts for a small fraction of the total spectral weight. This is consistent with a quasi-circular Fermi surface at $U = 0$ as seen on Fig. 1 b): Luttinger's theorem predicts that the Fermi surface's shape should not change much with interactions, as long as we are far from the Van Hove singularity, which here is located near $p = 0.75$ [72]. This is not true for non-circular Fermi surfaces [73]. We verified that the small leftovers of Fermi surface in the latter patch follow a $T^2$ scattering rate, which is reminiscent of a Fermi liquid.

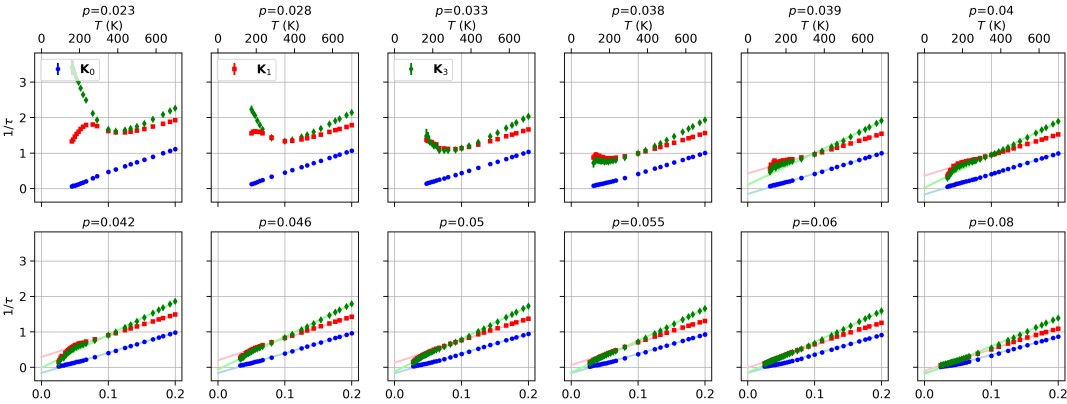

Figure 3: Scattering rate as a function of temperature for hole dopings between $p = 0.023$ and $p = 0.08$ at $U = 8.4$ for the zeroth, first and third patches of the triangular lattice in Fig. 1b). The temperature scale, on top, is fixed by taking $t = 0.3$eV, typical of the numbers for cuprates.

As stated earlier, $\omega/T$ scaling is usually associated with quantum criticality, but here we do not find it at $p^* \sim 0.06$. The procedure to check for $\omega/T$ is explained in Appendix D. We find, $\omega/T$ scaling only at $p = 0.04$ for $U = 8.4$, and at $p = 0.05$ for $U = 8.5$, at $T > 0.05$. For both values of $U$, this scaling is found only for patch 1 and 3 in a regime where the scattering rate is not linear in $T$ at low temperature. From Figs. 3 and 5 one can verify that the scattering rate at these two dopings is very similar. Indeed, in both cases, there is a downturn of the scattering rate around $T = 0.05$ for patch 1 and 3.

To clarify the origin of the quantum critical point and of $\omega/T$ scaling, Fig. 4 shows how doping varies as a function of chemical potential $\mu$ at $U = 8.4$ and $T = 0.05$. There is a first-order transition with coexistence between a pseudogap at $p = 0.02$ and a metal $p = 0.04$. This can be verified from the density of states computed on both sides of the phase transition with the maximum entropy method [74], as illustrated on the bottom row of the figure. The loss of spectral weight near the Fermi level is clear on the left plot while the quasiparticle peak is clear on the right plot [60]. There is also a first-order transition on the electron-doped side around $x = 0.02$, as shown on the top plot of Fig 4. The inset of that figure shows the local scattering rate $1/\tau(T)$ at $U = 8.4$ for both $x = 0.02$ and $p = 0.04$. On the electron-doped side, just like on the hole-doped side, there is a downturn in $1/\tau$ near $T = 0.05$. This suggests that this downturn in $1/\tau(T)$ is intrinsic to the proximity of the first-order transition.

In the case of the square lattice [37, 38], the analog of the first-order Sordi transition that we just discussed is continuously connected to the Mott transition. The first-order Sordi transition on the triangular lattice behaves similarly [39]. In particular, there should be a finite-temperature critical point. In addition, in single-site dynamical mean-field theory, the Mott transition has a quantum-critical point at the end of a coexistence region [15,75] leading us to suggest that the quantum critical point that we see at $p^* = 0.06$ on the triangular lattice has a similar origin.

Back to $\omega/T$ scaling. For both $p = 0.04$ and $x = 0.02$, there is range of $\omega/T$ scaling of the self-energy that breaks down at temperatures below $T \sim 0.05$ for $p = 0.04$, and below $T \sim 0.07$ for $x = 0.02$. These are the temperatures where the behaviour of the scattering rate in Fig. 4 changes drastically. For the hole-doped case, the critical point of the Sordi transition appears to be near $T = 0.05$ , while it seems to be at a slightly higher temperature for the electron-doped case.[3] Thus, the $\omega/T$ scaling appears to emerge from the finite-temperature

---

[3]It is known from Ref. [39] that the electron-doped hysteresis for the Sordi transition is far less sensitive to

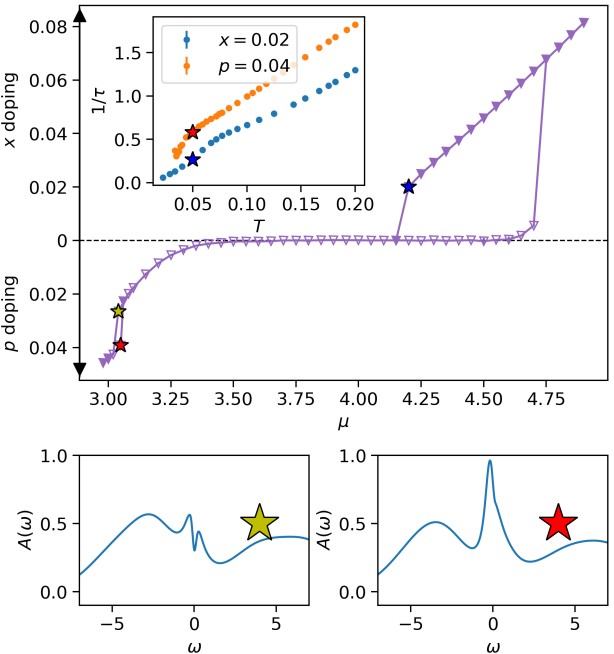

Figure 4: On top, electron doping $x$ and hole doping $p$ as a function of the chemical potential $\mu$ at $T = 1/20$ and $U = 8.4$. The inset presents the local scattering rate as a function of temperature for $p = 0.04$ (red star) and $x = 0.02$ (blue star). The scattering rate at $p = 0.04$ is larger than at $x = 0.02$. Above the position of the star, the temperature dependencies are similar. To illustrate the first-order transition, the bottom row of the plot shows the local density of states for the same chemical potential and two coexisting dopings, $p = 0.025$ and $p = 0.04$, showing a pseudogap in the first case and a quasi-particle peak in the second case.

critical point of the Sordi transition.

## 3.2 Interaction driven $T$-linear scattering rate

Before we discuss $T$-linear scattering, we point out that there is an unusual region in the high doping phase diagram Fig. 2b located between 0.15 and 0.2 doping. Indeed, there we find a $T^2$ dependence of the scattering rate, a result usually associated to a Fermi liquid. This result is surprising since Fermi liquids are usually found at higher dopings. One could think that this is caused by the odd number of electrons in the cluster when the doping is close to $p = 1/6$. Indeed, an odd number of electrons increases the entropy [76], which may push the Mott transition to larger $U$, as seen in Refs. [76] and [60]. However, a calculation of the scattering rate as a function of temperature at $p = 0.17$ in a four-site cluster led to the same $T^2$ dependence of the scattering rate at low temperature. This means that this $T^2$ regime found in the triangular lattice is not an artifact of the cluster used. We do not have an explanation for this Fermi liquid-like behaviour for a small range of dopings between 15% and 20%. A similar $T^2$ regime is found at comparable doping in cuprates, but in that case it appears to be due to Fermi-surface reconstruction from charge-density waves [77, 78].

changes of parameters than its hole-doped counterpart. As a consequence, we expect to find the critical point for the electron-doped Sordi transition at higher temperature.

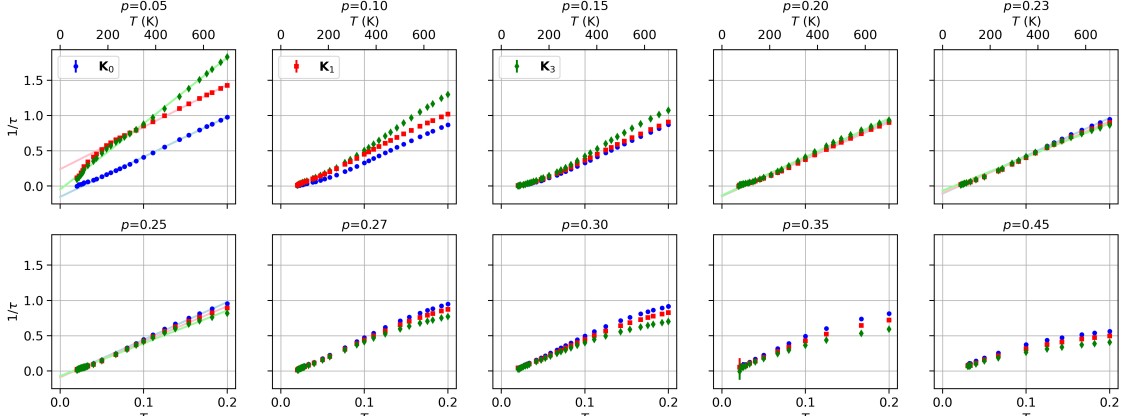

Figure 5: Scattering rate as a function of temperature for a large range of dopings at $U = 8.5$ for the zeroth, first and third patches of the triangular lattice in Fig. 1b). A linear fit on the scattering rate as a function of temperature is presented for hole doping between $p = 0.05$ and $p = 0.45$. For $p = 0.05$, the linear fit is done for $T > 0.2$. The temperature scale is fixed by taking $t = 0.3$eV, typical of the numbers for cuprates.

Let us move to $T$-linear scattering at large doping. It is present in Fig. 2b) below $T \sim 0.03$, for $p$ between 0.18 and 0.34. For higher temperatures, the exponent $n$ increases. It is remarkable that the slope of the $T$-linear scattering has been found experimentally [10] to satisfy the relation $\hbar/\tau = \alpha k_B T$ with $\alpha \sim 1$. Setting aside the difference between transport scattering time and single-particle scattering time, we note that the value of $\alpha$ is often found experimentally using the Drude formula.

$$\tau_{quasi} = \frac{m^*}{ne^2\rho}\,, \tag{8}$$

with $m^*$ instead of $m$. In that case the resulting scattering time is the quasiparticle scattering time [79]. In order to compare with our results then, the electron scattering rate $-\text{Im}\Sigma(\omega = 0)$ must be multiplied by the quasiparticle weight $Z$, which is obtained from the following relation [80]

$$Z = \left(1 - \left.\frac{\partial \text{Re}\Sigma(\omega)}{\partial \omega}\right|_{\omega \to 0}\right)^{-1} \tag{9}$$

$$\approx \left(1 - \left.\frac{\text{Im}\Sigma(\omega_n)}{\omega_n}\right|_{\omega_n = 0}\right)^{-1}. \tag{10}$$

The local quasiparticle scattering rate at $U = 8.5$ as a function of temperature is displayed in Fig. 6. The inset shows a clear linear temperature dependence at low temperature with $\alpha = 0.98 \pm 0.03$, very close to unity, similarly to the square lattice [12]. Thus, the interaction-driven $T$-linear scattering rate found in the triangular lattice also displays Planckian dissipation. Geometrical frustration then, does not seem to affect the value of $\alpha$ at high doping. Note that the value of $Z$ is about equal to 1/3 for the data in the inset of Fig. 6. The unrenormalized data is in Fig. 5.

Another characteristic of strange metals is that their self-energy scales with $\omega/T$ [12, 68]. This type of scaling is often related to quantum criticality. Here, we do not find $\omega/T$ scaling. This further asserts the idea that quantum criticality is not responsible for the Planckian dissipation that we see in the high-doping range. We further comment on scaling in Appendix D.

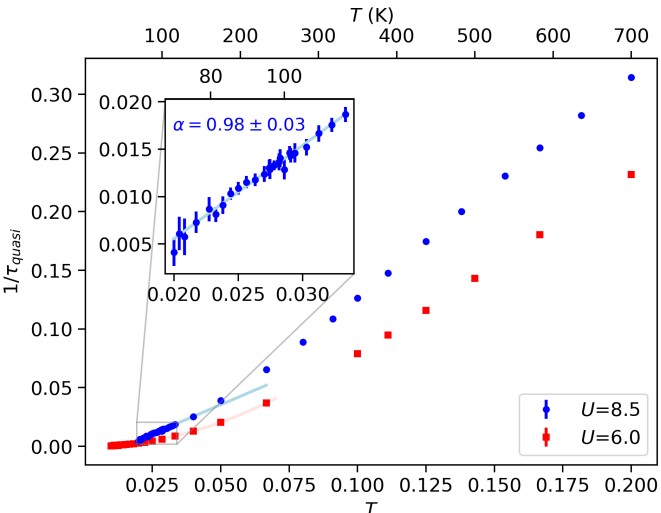

Figure 6: Local quasiparticle scattering rate as a function of temperature at $p = 25\%$ and $U = 8.5$. A linear fit is performed for temperature between $T = 0.02$ and $T = 0.03$ in the inset. The value of $\alpha$ obtained with this fit is presented in the inset. For $T > 1/15$, the slope $\alpha$ increases to $1.89 \pm 0.01$. On the other hand, the electron scattering rate $1/\tau$ has a slope $\alpha = 3.48 \pm 10$ for $T < 0.3$.

In order to find the origin of Planckian dissipation, the value of $U$ was lowered to see if it would survive. The scattering rates as a function of temperature at $p = 0.25$ for both $U = 6$ and $U = 8.5$ are presented on Fig. 6. We see that at lower $U$, the $T$-linear scattering rate is replaced by a $T^2$ scattering rate [81]. This could be expected from an increase in the coherence temperature when $U$ is decreased.

# 4 Discussion

After a discussion of our results on the triangular lattice, we compare with the square lattice results. Log-Log plots of the temperature dependence of the scattering rates may be found for a few dopings in Appendix E.

## 4.1 Two regions of linear in $T$ scattering rates on the triangular lattice

Research on the Hubbard model on the triangular lattice allows to discriminate the effect of long- vs short-range AFM fluctuations. Finding $T$-linear scattering rate in this model shows that only short-range fluctuations are important, particularly since many studies do not find magnetic ordering at half filling in the range of interaction strength we studied [40–46].

Strange metallicity is defined by $T$-linear scattering rate for $T \to 0$. It is usually associated to a quantum critical point at $p^*$ [68, 82]. Fig. 5 shows that linear fits of the scattering rate as a function of temperature extrapolate to negative values of $1/\tau$ at $T = 0$ for all dopings. Thus, $T$-linear scattering rate has to disappear at $T > 0$. The sign problem prevents us to go to low enough temperature to observe that.

As $U$ increases, the finite-temperature critical point of the Sordi transition moves to lower temperature [38]. It may eventually reach zero temperature, in which case it would turn into a quantum critical point and there would be no downturn of the scattering rate. The scattering rate $1/\tau$ would likely extend all the way to $T \to 0$. An analogous quantum-critical point is found at $T = 0$ in the two orbital Hubbard model with Hund coupling [83].

That the interaction-driven $T$-linear scattering rate is found for a wide range of dopings, $0.18 < p < 0.34$, suggests that it does not emerge from a quantum-critical point. This is supported by the lack of $\omega/T$ scaling. The extrapolation of the linear behavior to negative temperatures at $T = 0$ suggests instead a crossover from linear to Fermi liquid $T^2$ at a temperature lower than what is computationally achievable with DCA. Such a crossover is visible at $U = 6$ in Fig. 6. The crossover temperature decreases as $U$ increases.

Note that the interaction-driven $T$-linear scattering rate that we find here is similar to what is found on the 8-site square lattice with DCA where, however, $\omega/T$ scaling was found at one doping and connected to the effect of spin fluctuations [12].

Since $\omega/T$ scaling is not found in the interaction-driven $T$-linear scattering rate, we look for other possible scalings. We find in Appendix D that $\text{Im}(\Sigma(i\omega_n), \mathbf{K}_3)/(\text{Im}\Sigma(i\omega_n = 0, \mathbf{K}_3))$ scales like $\omega/T^z$, where $z$ varies between 2 and 2.3 depending on the doping and of $\mathbf{K}_i$. This type of scaling of the self-energy is different from what is expected from both Fermi liquid theory, where $-\text{Im}\Sigma \sim \omega^2 + T^2$, and from quantum-critical strange metals. The scaling encountered in this $T$-linear scattering rate region is also dimensionful, which means that it is non-universal. We do not have any explanation for this type of scaling.

The contrasting temperature dependence of the scattering rates on the different patches and their relation to the pseudogap is discussed in Appendix F.

It is well known that there is a critical point associated to the Mott transition at half-filling in single site DMFT [84, 85]. In the bad-metal regime at high temperature, a linear in $T$ scattering rate is also found [16, 17, 86]. It is also known that the Mott critical point has an influence away from half-filling [15]. To verify if single site DMFT at finite doping also has an interaction-driven $T$-linear scattering regime at large doping, calculations as a function of temperature at $p = 0.25$ for cluster sizes $N_c = 1$, $N_c = 2$ and $N_c = 4$ were performed, and are displayed in Fig. 10 of Appendix A. They show that while single site DMFT calculations lead to $T^2$ scattering rate at low temperature, clusters with $N_c \geq 2$ lead to $T$-linear behaviour at low temperature. This strongly suggests that superexchange is crucial for the $T$-linear behaviour of the scattering rate in this regime.

Finally, contrary to claims found in the literature [87], we do not find a link between Planckian dissipation and quantum criticality. Indeed, the interaction-driven regime, that displays Planckian dissipation, does not display quantum critical scaling.

## 4.2 Comparison with the square lattice

The results for the local scattering rate as a function of temperature on the square and the triangular lattices are presented in Fig. 7 for $p = 0.25$ and $U = 8.5$. We find that the effect of geometrical frustration does not affect the behaviour of the scattering rate at low temperature. This leads us to believe that the underlying physics responsible for the $T$-linear scattering rate is the same for both the square and triangular lattices. Based on this assumption, we compare our results for the triangular lattice to the characteristics of the strange metal phase found in cuprates.

We saw that the interaction-driven $T$-linear scattering rate is lost at temperatures higher than $T \sim 0.03$ where the exponent of the $T$ dependence increases. This kind of deviation from $T$-linear scattering rate is commonly found in LCCO, PCCO, Nd-LSCO and Bi2212 [22, 88]. Even though a direct comparison with cuprates is not warranted, we mention that the temperature at which the $T$-linear scattering rate is lost on the triangular lattice ($T \sim 150K$) is similar to what is found in Nd-LSCO and Bi2212 ($T \sim 120K$) [22].

Also, as in cuprates, the scattering rate in the interaction-driven $T$-linear scattering rate region is near isotropic. Note that the cuprate measurements, however, were done close to a van Hove singularity [23].

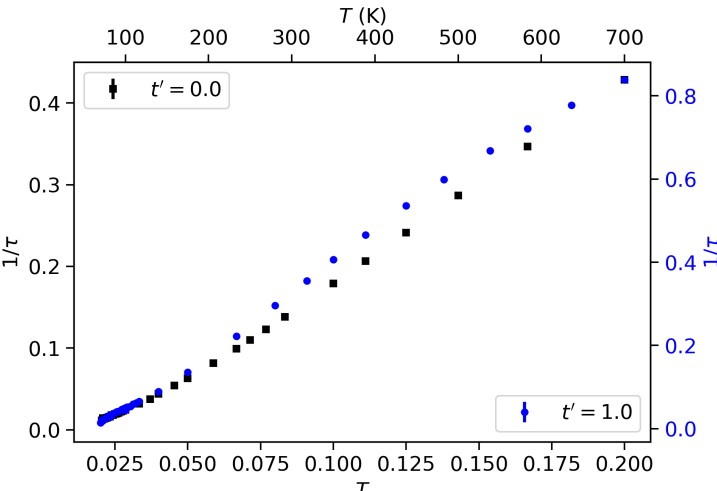

Figure 7: Local scattering rate as a function of temperature for both the square lattice ($t' = 0$) and the triangular lattice ($t' = -t = 1$) for $p = 0.25$ and $U = 8.5$. The temperature scale is fixed by taking $t = 0.3$eV, typical of the numbers for cuprates.

Another similarity between the strange metal found in cuprates and the interaction-driven $T$-linear scattering rate is that both display Planckian dissipation. This means that geometrical frustration does not change the slope. Moreover, Fig. 6 shows that $T$-linear scattering rate at small $U$ either disappears completely or appears at smaller temperature, which would be surprising given the negative intercept of the extrapolated $T = 0$ result. Thus, Planckian dissipation occurs when interactions are sufficiently strong, with no other obvious explanation.

There are important differences between what is found here on the triangular lattice and what is found in cuprates like LSCO. Cuprates have a $T$-linear scattering rate on a wide range of dopings like we find, but linearity extends down to $T \to 0$ on the entire range of dopings [10]. Moreover, they exhibit $\omega/T$ scaling for dopings away from $p^*$ [89]. Nevertheless, the similarity between our phase diagram Fig. 2b) and Figure 1 of Ref. [90] is remarkable.

In real materials like cuprates, the effect of disorder may be important for observing linear in $T$ scattering rate, as emphasized in quantum-critical models [91, 92], in SYK models [93] or much earlier in Boltzmann transport [94, 95]. In the latter case, Rosch [94] pointed out that disorder may invalidate the Hlubina-Rice argument [95] that Fermi-liquid like regions of the Fermi surface with $T^2$ scattering rate would short-circuit hot-spots with $T$ scattering rate. With disorder, the Hlubina-Rice argument can indeed be invalid. Using a caricature to account for an elastic scattering rate $\Gamma_0$ with Mathiessen's rule, one finds that as $T$ approaches zero, $\Gamma_0$ becomes larger than $T^2$ faster than it becomes larger than $T$. The effect of disorder on the scattering rate remains to be studied with DCA.

# 5 Conclusion

We used DCA with a CT-AUX continuous-time impurity solver to study $T$-linear scattering rate in the hole-doped triangular-lattice Hubbard model. We find that the phase diagram displays two metallic regions with linear in $T$ scattering rates. The first one, that we call Mott-driven, is found for low dopings near the Sordi transition. This $T$-linear scattering rate has and emerges from a single doping $p^* \sim 0.06$ and is very close to the $\omega/T$ scaling at $p \sim 0.04$.

The second $T$-linear scattering rate region, that we call interaction-driven $T$-linear scattering rate, has no $\omega/T$ scaling and is found for a wide range of dopings. It does not emerge from a quantum-critical point. The linear fits of the scattering rate as a function of temperature extrapolate to negative value of $1/\tau$ at $T = 0$, which suggests a crossover to a Fermi liquid regime at a temperature lower than what is actually possible to achieve because of the sign problem. Although there is no identifiable quantum critical point, we found Planckian dissipation in this regime of interaction-driven $T$-linear scattering rate at $p = 0.25$. We also showed that clusters of at least two sites are necessary to observe linear in $T$ scattering rate at low temperature in this interaction-driven regime, strongly suggesting that superexchange is crucial to observe this behavior.

The similarities and differences between the strange metal and the two linear in $T$ regimes that we identified are summarized in Table 1.

This study is the first to report that there might be two different regimes for $T$-linear scattering in strongly correlated materials. This discovery may have an impact on our current understanding of strongly correlated materials, and more particularly, could impact our vision of the strange metal in cuprates. It would thus be important to verify if other models, or calculation technique, find $T$-linear scattering rate without quantum criticality and phonon interactions or near the Sordi transition.

## Acknowledgments

Useful discussions with A. George, O. Gingras, P.A. Graham, G. Grissonnanche, C. Proust, G. Sordi and L. Taillefer are acknowledged.

**Funding information**    This work has been supported by the Natural Sciences and Engineering Research Council of Canada (NSERC) under grant RGPIN-2019-05312 and by the Canada First Research Excellence Fund. The computational resources were provided by Calcul-Québec and the Digital Research Alliance of Canada.

## A    Dependence on $N_c$

To verify the accuracy of our results, the scattering rate as a function of temperature for a 12 site bipartite cluster was computed with DCA for the two values $U = 8.4$ and $U = 8.5$. Recall, as explained in Fig. 1, that by bipartite we mean that the cluster would be bipartite if we were to take $t' = 0$. The results obtained for $U = 8.5$ are presented at Fig. 8. The 12-site bipartite-cluster results are very similar to those of the 6-site cluster for large dopings. At smaller dopings, it is not the case anymore.

One can understand why by looking at Fig. 9. Results for the Widom line [96] indicate that the Mott transition is at larger $U$ in the 12-site cluster. With the Mott transition for the 12-site cluster at much larger $U$ [39], effects from the Sordi transition on the scattering rate do not appear at low doping. Hence, we should not expect results at low dopings to be the same for both of those clusters. Furthermore, because of the sign problem, it is impossible to get accurate results below $\beta = 11$ on the 12-site cluster, which means that it is not possible to verify our results at the lowest temperatures.

The scattering rate as a function of temperature in the interaction-driven regime, as well as in the Fermi liquid regime found at $p = 0.17$, is also displayed for clusters of size $N_c = 6$ and smaller on Fig. 10. The motivation for the use of the $N_c = 4$ cluster at $p = 0.17$ was to verify if the Fermi liquid regime found in the $N_c = 6$ cluster was due to the fact that near a doping of $p = 5/6$, the average number of electrons is odd. We find that even in the $N_c = 4$

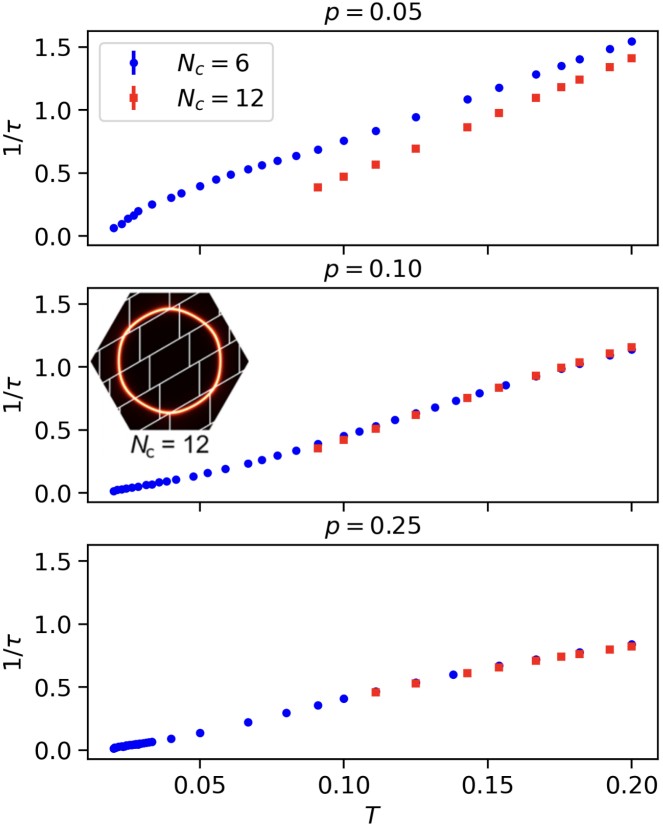

Figure 8: Local scattering rate as a function of temperature for $U = 8.5$ and three dopings, $p = 0.05$, $p = 0.1$ and $p = 0.25$, for the six-site bipartite cluster in blue and the twelve-site bipartite cluster in red. In the middle subplot, the subdivision of the Brillouin zone into the 12 different patches is given with the non-interacting Fermi surface at half-filling.

cluster, a Fermi liquid is found at low temperature. This means that this Fermi liquid region is not an artifact of the cluster used.

At $p = 0.25$, Fig. 10 shows that in the single-site cluster, the scattering rate becomes quadratic at low temperature. However, the $T$-linear regime is retrieved at low temperature for $N_c \geq 2$ clusters. This means that short-range interactions, such as superexchange, are important in order to find the interaction-driven $T$-linear scattering rate regime.

# B Polynomial fit on first Matsubara frequencies

Much of the literature uses a polynomial fit on the first Matsubara frequencies to find the approximate value of a given observable at $\omega = 0$. This is usually a good approximation [12]. Testing and comparing low frequency results from such techniques, we conclude that the best polynomial fit is of order three.

We also compared with maximum-entropy analytic continuation [74] and with a second degree least-square regression on the first six Matsubara frequencies. The results for the second degree least-square regression are in concordance with the second degree polynomial fit. Although the results at order 4 better fit the maximum-entropy technique, as seen on Fig. 11, it is sensitive to small errors in the input observables. Since the shape of the final fit does not change much, this indicates that the results given in the article are valid.

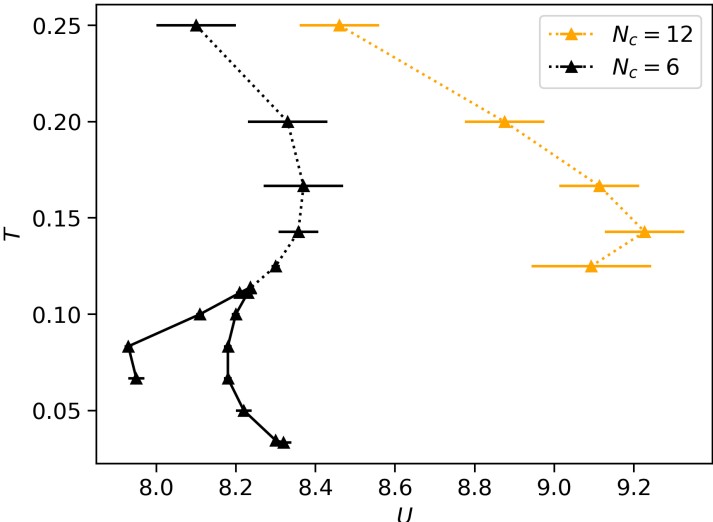

Figure 9: Mott transition and Widom line for six-site and twelve-site clusters of the triangular lattice. The dotted line corresponds to the Widom line [96], a crossover. The solid lines correspond to $U_{c1}$ and $U_{c2}$ for the Mott transitions.

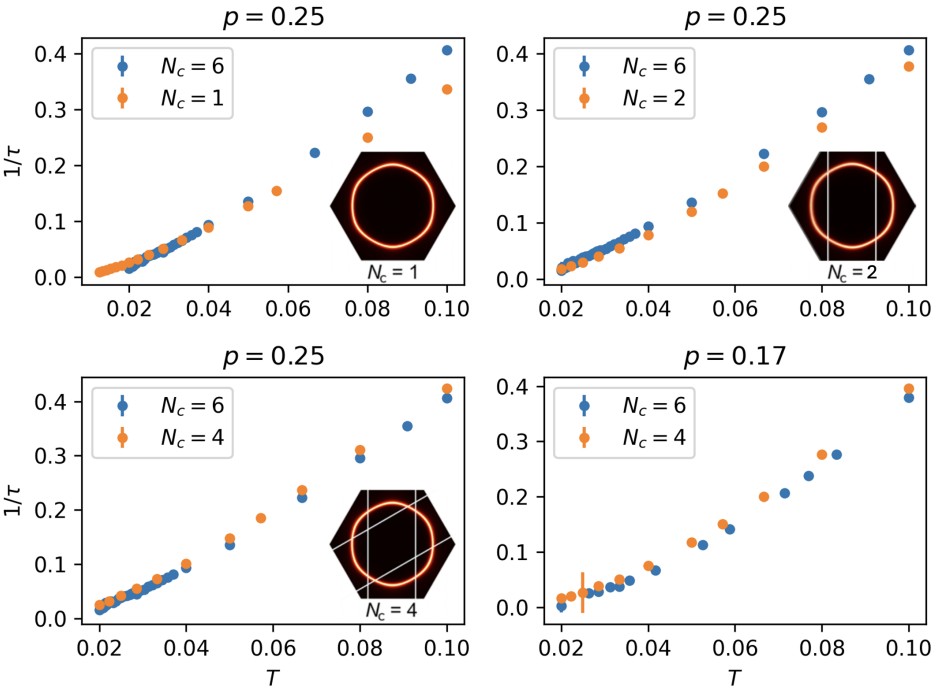

Figure 10: Local scattering rate as a function of temperature for different clusters, at $p = 0.25$ and $p = 0.17$ for $U = 8.5$. One needs at least $N_c = 2$ to find a linear regime.

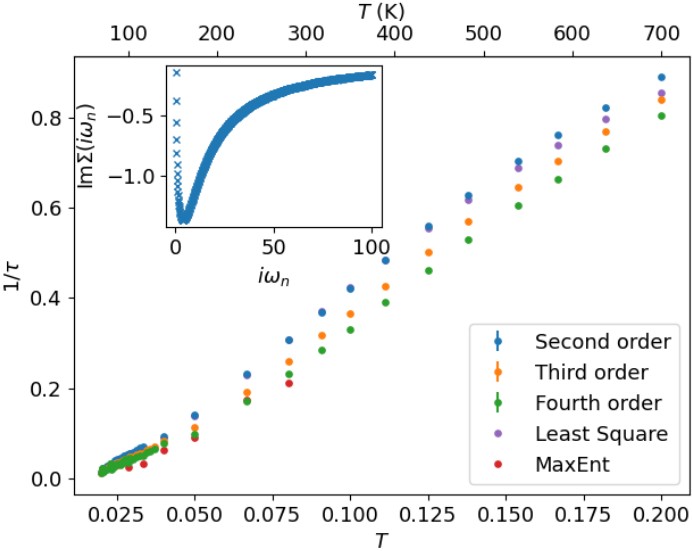

Figure 11: Local scattering rate ($-\text{Im}\Sigma(\omega = 0)$) as a function of temperature $T$ for $p = 0.25$ and $U = 8.5$ obtained from the Matsubara self-energy using polynomial fits of different orders. Also shown is a second order least-square regression on the first six Matsubara frequencies and $\text{Im}\Sigma(\omega = 0)$ obtained with the MaxEnt method OmegaMaxEnt [74]. The inset displays the imaginary part of the self-energy as a function of the Matsubara frequencies for $T = 0.02$, $p = 0.25$ and $U = 8.5$

## B.1 Planckian dissipation

The slope of the scattering rate as a function of temperature decreases when the order of the polynomial fit increases. To verify if the quasiparticle scattering rate is still Planckian with higher-order polynomial fits of the self-energy, the slope of the quasiparticle scattering rate as a function of temperature was computed for polynomial fits of higher order. We find slopes $\alpha = 0.87 \pm 0.04$ and $\alpha = 0.78 \pm 0.02$ with polynomial fits of order four and five respectively. These values of $\alpha$ are still within the slope found experimentally in materials displaying Planckian behaviour [22]. Thus, even if the slope of the quasiparticle scattering rate depends on the order of the polynomial fit on the Matsubara frequencies, we find that the $\alpha$ obtained remains close to the Planckian limit of $\alpha = 1$ in the interaction-driven $T$-linear scattering rate.

One should note that the results at $\omega = 0$ from the maximum entropy analytic continuation are not very stable in temperature, so we did not push our analysis further for this.

## C Phase diagram

In order to strengthen the link between the finite-doping continuation of the Mott transition, the Sordi transition [38], and the low-doping $T$-linear regime, the doping $p$ was computed as a function of chemical potential at fixed temperature and $U = 8.4$. Fig. 12 presents an improved phase diagram at $U = 8.4$ and low dopings, where the Sordi transition and the Widom line are added to the colorplot in Fig. 2.

To obtain the phase diagrams on Fig. 2, the scattering rate as a function of temperature for the different dopings were fitted using Legendre polynomials of degree 7. Then, a fit of the form $aT^n + b$ was performed on each group of 10 points of the Legendre fit to obtain the local value of $n$ as a function of temperature and doping. The values of $n$ were then interpolated on a meshgrid to obtain Fig. 2. The non-interpolated values of $n$ are presented

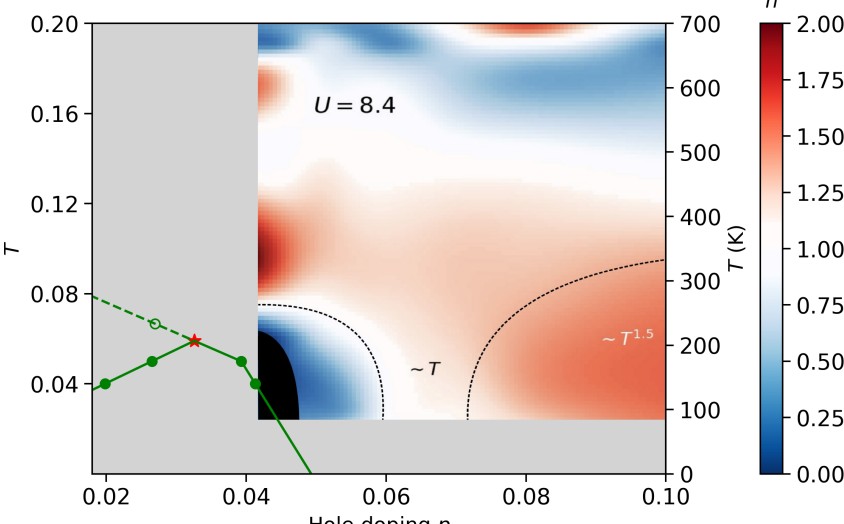

Figure 12: Phase diagram of the triangular lattice Hubbard model at $U = 8.4$. In addition to the colorplot from Fig. 2a), the Sordi transition and the Widom line between the PG and cFL are also displayed, respectively, with a full line and a dotted line. No value of the exponent $n$ of the scattering rate as a function of temperature was computed in the dark region near $p = 0.04$, as seen in Fig. 13.

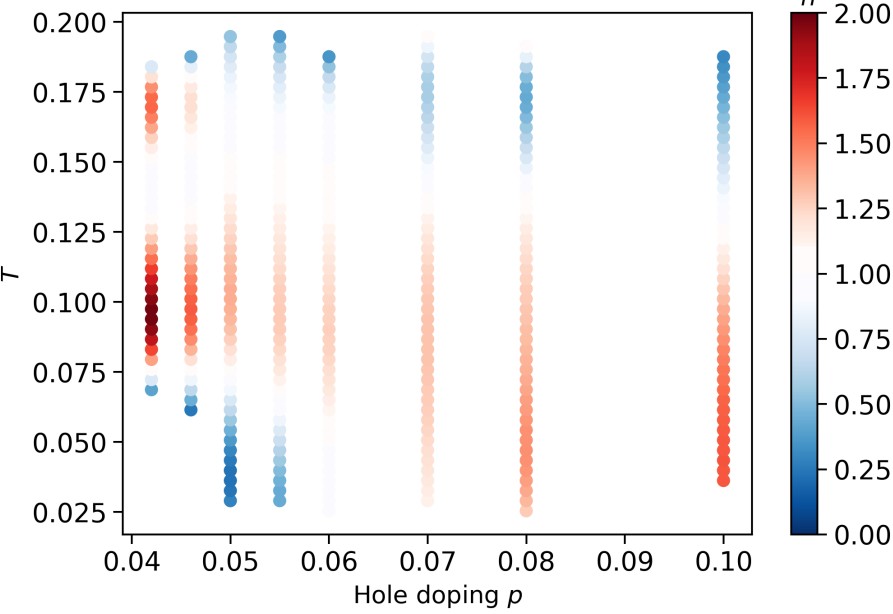

Figure 13: Raw data for the temperature-doping phase diagram Fig. 2a) of the local scattering rate for $U = 8.4$. The value of $n$ obtained from a fit of the form $1/\tau = \alpha T^n + b$ of the local scattering rate is color coded and interpolated to obtain Fig. 2a).

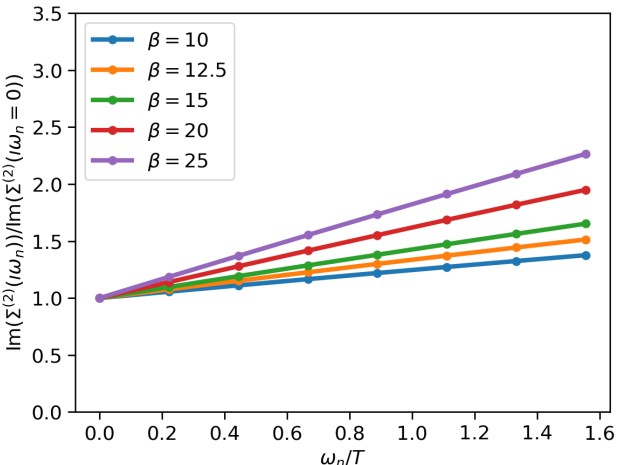

Figure 14: $\text{Im}\left(\Sigma(i\omega_n), \mathbf{K}_1\right)/\left(\text{Im}\Sigma(i\omega_n = 0), \mathbf{K}_1\right)$ as a function of $\omega_n/T$ for temperatures where a T-linear scattering rate is found at $p = 0.25$. Only the small values of $\omega_n/T$ are displayed in order to verify that $\omega/T$ scaling does not hold.

on Fig. 13. At low dopings and low temperature, the scattering rate could not be fitted with the form $aT^n + b$, hence the absence of points. Different orders of the Legendre polynomial and number of points for the fits were tested to make sure that the values of $n$ color coded on the figure were independent of these parameters.

## D  $\omega/T$ scaling

Most strange metals have an optical conductivity that scales like $\omega/T$ so in our case we expect a self-energy that has the form $-\text{Im}\Sigma(i\omega_n, T) = \lambda T^\nu \Phi(\frac{i\omega_n}{T})$, where $\lambda$ is some constant and $\Phi$ is a function of $i\omega_n/T$ [35,68]. The $\omega/T$ scaling is then obtained from analytic continuation $i\omega_n \to \omega + i\eta$. This type of scaling is normally associated with quantum-critical points.

We can verify whether our data follows $i\omega_n/T$ scaling by computing

$$\text{Im}\Sigma(i\omega_n, T)/\text{Im}\Sigma(i\omega_n = 0, T),$$

that should then scale as $\Phi(\frac{i\omega_n}{T})/\Phi(0)$. There are two regimes of doping with linear in $T$ scattering. Let us begin with the large doping regime. There is T-linear scattering rate for $p = 0.25$ and $T < \frac{1}{33}$. The above ratio as a function of $i\omega_n/T$ is presented on Fig. 14 for the first patch. We find that the self-energy for all patches in this interaction-driven $T$-linear scattering rate does not display $\omega/T$ scaling. The absence of $\omega/T$ scaling, along with the existence of Planckian dissipation for a large range of dopings, leads us to conclude that $T$-linear scattering here does not emerge from quantum criticality.

There is however a finite temperature critical point at $p = 0.04$ for $U = 8.4$.

$$\Sigma(i\omega_n, T)/\Sigma(i\omega_n = 0, T),$$

as a function of $i\omega_n/T$ for this doping is presented at Fig. 15. We see that, for temperatures higher than the finite-temperature of the critical point, the Mott-driven $T$-linear scattering rate displays $\omega/T$ scaling. To find out whether there is a different scaling of the self-energy in the interaction-driven $T$-linear scattering we computed $\Sigma(i\omega_n, T)/\Sigma(i\omega_n = 0, T)$ as a function of $\omega/T^z$. The value of $z$ was varied until each temperature has the same scaling of

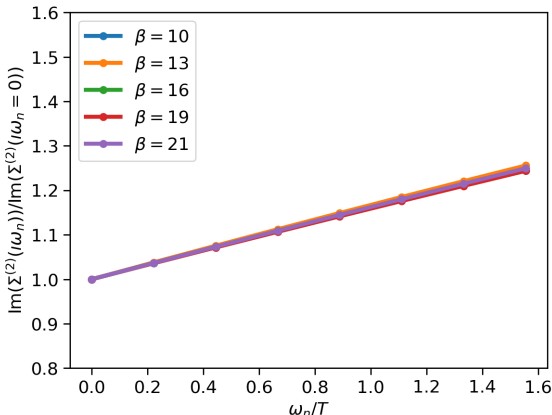

Figure 15: $\text{Im}\left(\Sigma(i\omega_n, \mathbf{K}_1)\right) / \left(\text{Im}\Sigma(i\omega_n = 0, \mathbf{K}_1)\right)$ as a function of $\omega_n/T$ for temperatures where a T-linear scattering rate is found at $p = 0.04$ and $U = 8.4$. Only the small values of $\omega_n/T$ are displayed in order to verify the $\omega/T$ scaling.

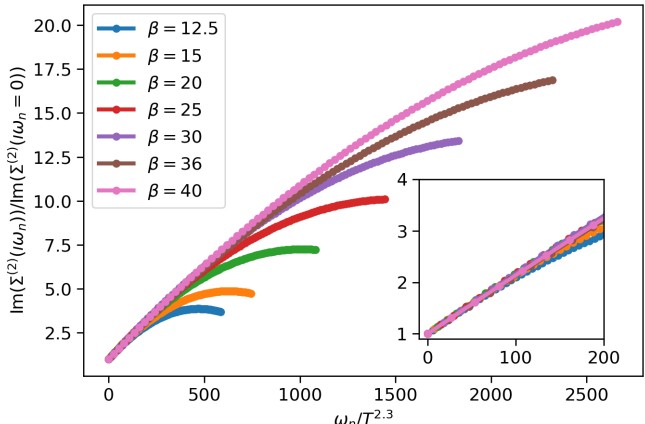

Figure 16: $\text{Im}\left(\Sigma(i\omega_n), \mathbf{K}_3\right) / \left(\text{Im}\Sigma(i\omega_n = 0, \mathbf{K}_3)\right)$ as a function of $\omega_n/T^{2.3}$ for different temperatures for doping $p = 0.25$. The insert shows the low-temperature scaling.

$\Sigma(i\omega_n, T)/\Sigma(i\omega_n = 0, T)$ at low Matsubara frequency. We find $\omega/T^{2.3}$ scaling, as shown in Fig. 16.

# E Scattering rate

In order to highlight the different scattering-rate regimes on the triangular lattice, the results as a function of temperature are also displayed on a log-log scale on Fig. 17. In the Mott-driven $T$-linear regime at small dopings, the low-temperature scattering rate for the different patches changes drastically with doping, as shown on Fig. 18.

Finally, the quasiparticle scattering rate at $p = 0.06$ is presented in Fig. 19 to support our claim that the Mott-driven $T$-linear scattering rate regime has a slope different from unity, hence it does not strictly-speaking display Planckian dissipation.

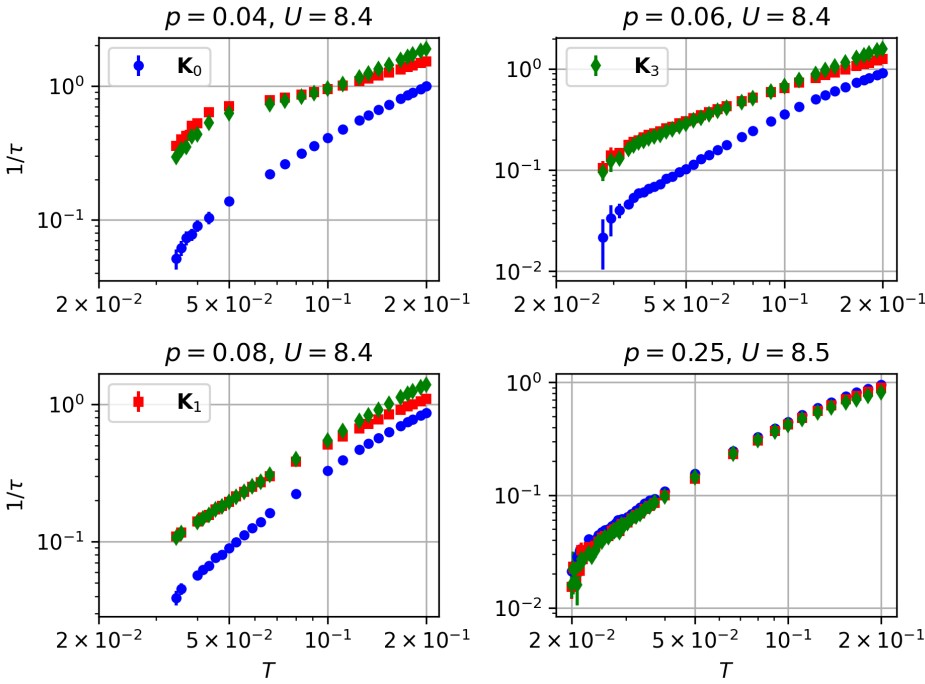

Figure 17: Scattering rate as a function of temperature on a logarithmic scale for different regimes displaying $T$-linear scattering rate. The different colors refer to different patches.

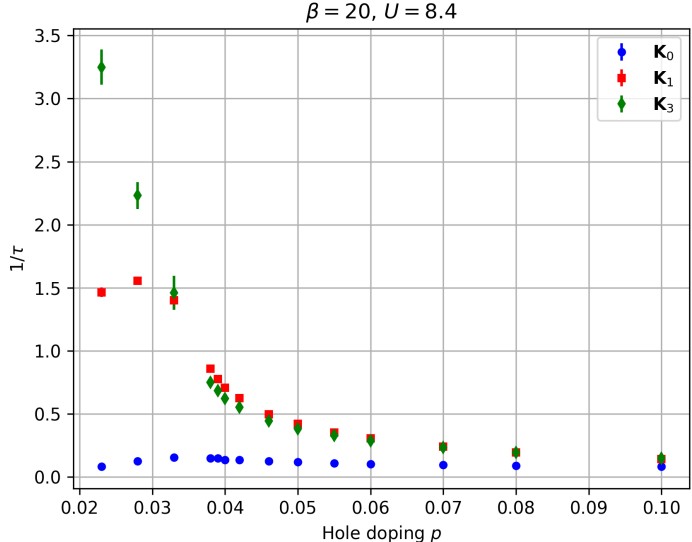

Figure 18: Scattering rate as a function of hole doping at $U = 8.4$ and fixed temperature $T = 1/20$, for patches $K_0$, $K_1$ and $K_3$.

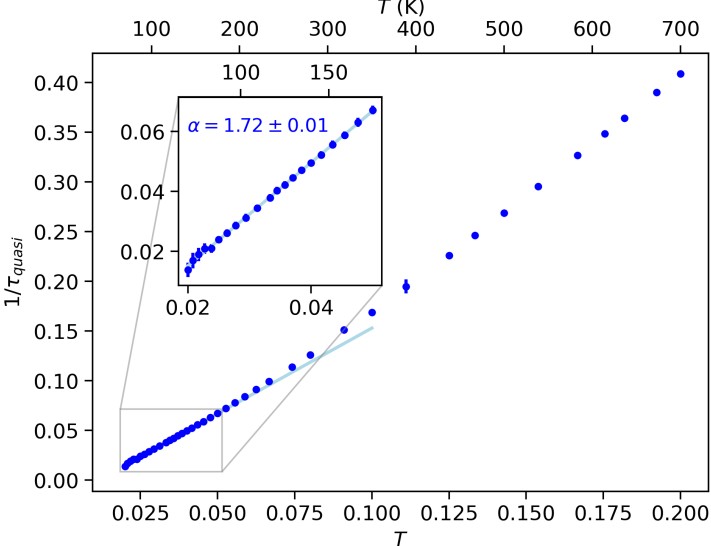

Figure 19: Quasiparticle scattering rate as a function of temperature for $p = 0.06$ and $U = 8.4$. A linear fit for the data at $T < 1/20$ is performed to obtain the value of $\alpha$.

## F    Spectral weight

In order to demonstrate the existence of a pseudogap in the triangular lattice, the spectral weight $A(K_i, \omega = 0)$ as a function of temperature is computed for different dopings. This data is presented on Fig. 20. We observe that for $p = 0.04$, $p = 0.06$ and $p = 0.25$, the spectral weight increases as the temperature decreases. This means that a quasiparticle peak is found at these points, indicating that there is no pseudogap. Thus, the downturn of the scattering rate for temperature below the critical point at $p = 0.04$ is not directly caused by a pseudogap, but is instead a precursor. However, we see that when the doping decreases to $p = 0.023$, we eventually see a decrease of the spectral weight at low temperature, which indicates that there is a pseudogap.

We also computed the density of state for different temperatures at $p = 0.25$, for $U = 6.0$ and $U = 8.5$. This data is presented on Fig. 21. We find that, just like in the Mott-driven regime, the density of state presents a peak near $\omega = 0$. Moreover, the increase of $U$ does not seem to qualitatively change the behaviour of the DOS. There are quantitative differences, such as the increase of the weight around $\omega = 10$ when $U$ increases.

Fig. 21 also presents the data for the quasiparticle weight $Z$ at $p = 0.25$ for both $U = 6.0$ and $U = 8.5$. We find that for $U = 6.0$, $Z$ becomes constant at the temperature where the scattering rate becomes $T^2$ in Fig. 6. This is expected from a Fermi liquid. However, for the interaction-driven regime at $U = 8.5$, $Z$ does not seem to become constant at low temperature. There seems to be an increase of $Z$ at the temperature where the scattering rate becomes linear, however, the large statistical errors at low $T$ prevents us from making clear statements.

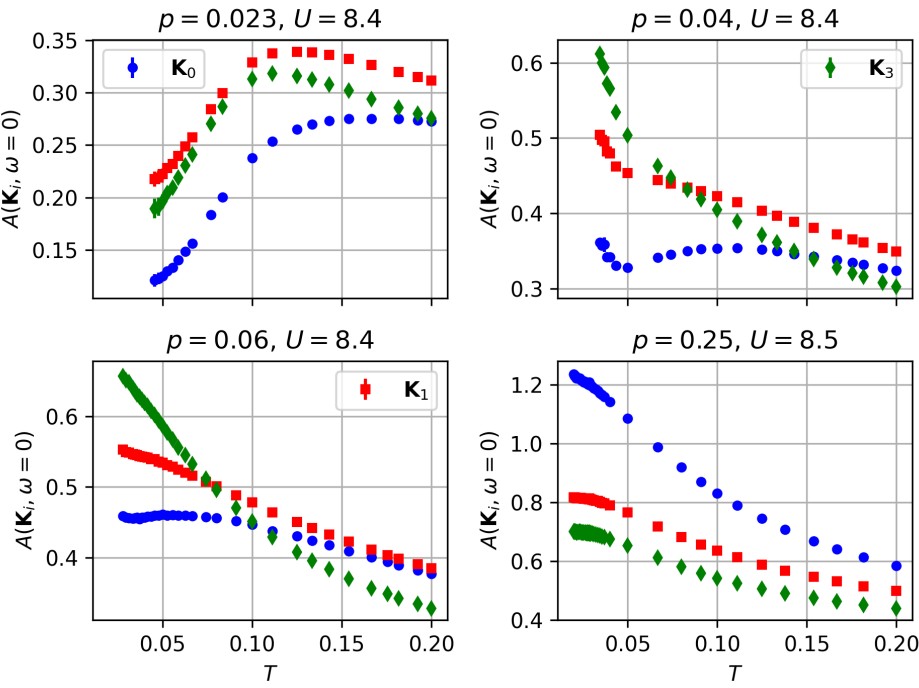

Figure 20: Spectral weight as a function of temperature for the patches $K_0$, $K_1$ and $K_3$ at different dopings and $U$.

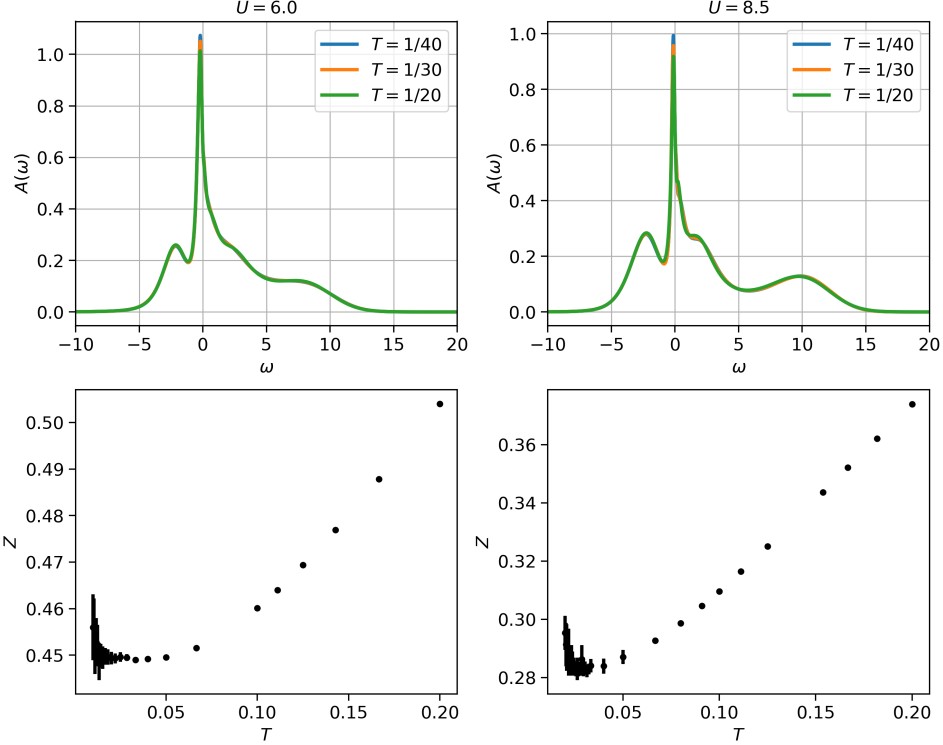

Figure 21: (Top) Density of state for $p = 0.25$ at $T = 1/20$, $T = 1/30$ and $T = 1/41$ obtained from the maximum entropy method, for $U = 6.0$ and $U = 8.5$. (Bottom) Quasiparticle weight $Z$ as a function of temperature for $p = 0.25$ and both $U = 6.0$ and $U = 8.5$.

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
