# Peer review of "Two $T$-linear scattering-rate regimes in the triangular lattice Hubbard model"

_SciPost Physics, doi:SciPost Phys. 17, 072 (2024)_

## Round 1 · Referee Report · Anonymous (Referee 1) · 2024-1-5

Strengths
- Timeliness of the topic
- Rare parameter window to be reach by numerical methods
- Careful and balanced writing
Weaknesses
-
Disconnect between the studied model (triangular lattice) and the actual experimentally relevant playgrounds (cuprate bethe lattice)
-
Uncontrolled systematic errors stemming from the employed numerical method
-
Insufficient parameter window to address a sufficient part of the expected phenomenology
Report
The topic is highly important and promises a yet to be discovered universality of strongly correlated electron systems transcending quantum critical point or strange metal phenomenology. There are only few methods capable of even addressing such a parametric regime in the first place, even though it must be said that cluster methods are, to a large extent, uncontrolled as the interaction scale vs kinematic scale is chosen to locate in the intermediate to strong coupling regime.
I believe the paper to be relevant and worth publishing. What I do not understand is the insistence of connecting the triangular lattice Hubbard model, which is what the authors analyze, to the cuprates. I find these passages kind of a stretch and a bit arbitrarily assembled and logically synthesized. In particular, the authors should not forget the linear T scaling in ruthenate bilayers and other systems, which I would regard to be of similar relevance as the linear T behaviour in cuprates. My reading is that the window of applicability of their cluster methods, bound below through the sign problem and bound above through the analytic continuation needed to appropriately obtain the scattering rate from the self energy, is small, so they want to enforce some kind of scale similarity to any linear T candidate compound out there - and that seems to be the cuprates. I understand the strategy, but I would not follow it. Arguing this way assumes axiomatically that linear T behaviour be something so universal and so deeply founded in principal physics and not tied to specifics of a given material that one would dare to describe a linear T behaviour on a cuprate Bethe/square lattice with a Hubbard model on a triangular lattice. In the end, I believe this would lead to some kind of circular reasoning.
Requested changes
-Table 1 is misleading. Assign the top row the left label "Strange metal" - then the caption will match the Table.
-I suggest to more clearly organize the limiting factors of the method combined and in a separate subsection. I improved the reading of the manuscript.
-Similarly I suggest to make another subsection on a coherent and condensed argueing what one employs the triangular lattice Hubbard model in trying to explain linear -T behaviour on a square lattice.
The full reply is in the attached file.

Author: Jérôme Fournier on 2024-07-08 [id 4609]
(in reply to Report 3 by Thomas Schäfer on 2024-01-22)The full reply is in the attached file.
Attachment:
Two_mechanism_report_3-2.pdf

---

## Round 1 · Referee Report · Anonymous (Referee 2) · 2024-1-15

Strengths
- The identification of two distinct regions of T-linear scattering rate attributed to different physical mechanisms (the Sordi transition at low doping and strong interactions at higher doping) contributes valuable knowledge to the field.
2.The temperature-doping phase diagram provided is detailed and adds depth to the understanding of the triangular lattice Hubbard model.
Weaknesses
-
Model Justification : The manuscript does not sufficiently justify the choice and motivation of employing anisotropic hopping parameters t and t' for simulating hole doping. This introduces anisotropy into the lattice, potentially complicating the interpretation of the results (Major Concern 1).
-
Phase Diagram Clarity: The region claimed to exhibit T-linear behavior in the phase diagram is overly broad, making the evaluation of the boundaries unclear (Major Concern 2).
Report
by Jérôme Fournier, Pierre-Olivier Downey, Charles-David Hébert, Maxime Charlebois, André-Marie Tremblay
In this work, the authors investigate T-linear scattering rate in the triangular lattice Hubbard model obtained using the dynamical cluster approximation for six-site cluster. They clarified the temperature-doping phase diagram, and they find two regions of T-linear scattering rate at low doping region and larger doping region. The former is driven by Sordi transition and the latter caused by the strong interactions.
While the content is suitable for publication in SciPost Physics and demonstrates significant novelty, there are crucial concerns that must be addressed to ensure the clarity and rigor of the manuscript:
1. The authors introduce the triangular Hubbard model with two distinct types of nearest neighbor hopping parameters, t and t’ to simulate hole doping. This approach inherently introduces anisotropy into the lattice. While this is an interesting method, it could potentially complicate the interpretation of the results due to the introduced anisotropy. A more straightforward method to simulate hole doping, which avoids anisotropic effects, is to use an isotropic Hubbard model and adjust the chemical potential accordingly. It would be highly recommendable if the authors provide further clarification on their motivation for choosing this particular model with anisotropic hopping parameters. Understanding the specific advantages or the intended insights this method provides over the conventional approach of altering the chemical potential would be valuable. It would help in comprehensively assessing the implications of the results and the uniqueness of the application of this model in this context.
2. Concerning the phase diagram, the region claimed to exhibit T-linear behavior seems too broad, and the evaluation of the boundary is unclear. The manuscript states, “There is a slight deviation from the T-linear regime seen in Fig. 2a) for T ~ 0.08. The raw data for the scattering rate at p=0.06 in Fig.3 shows that this deviation from T-linearity is barely noticeable (at page 4, right column)." However, this deviation is not easily discernible, potentially due to the relatively small size of the figure and the fact that it is not plotted on a logarithmic scale, which may obscure deviations from T-linearity. To address this concern and clarify the observations, it would be prudent to include a version of Fig. 3 plotted on a logarithmic scale. Additional it would be helpful how authors determined the boundary of T-linearity in the phase diagram.
While the innovative contributions of this work are clear, I must emphasize that concern number 2 regarding the phase diagram boundaries is particularly critical. The broad definition of the T-linear regime warrants a more precise analysis, as it is fundamental to the paper's claims and conclusions.
Minor concern:
[Contents-Related Concerns]
1. The explanations for \tilde{K} and the six patches K_i are insufficient. An explanation should be added at the end of the first paragraph in Section IIB, where Fig. 1b is mentioned, to enhance readability for the reader.
2. Regarding “For this reason, we limit ourselves to T lower than 0.2” on page 3 right column. It is not very clear since the energy unit is not clearly defined.
3. To facilitate a clearer understanding of the manuscript, I recommend that the hole doping parameters p and x be thoroughly defined and explained prior to Sec. III. This would ensure that readers have a solid grasp of these key concepts before delving into the subsequent analysis and discussion.
[Typesetting and Formatting Concerns]
4. I have noticed some typographical errors in the manuscript that need addressing. For instance, in the first paragraph of Sec. IIC, "he" should be corrected to "the". Additionally, there is a reference error on page 4 in the right column where "Table II C" is mentioned; it should instead refer to "Table I". This should be amended to maintain accuracy in the document's cross-referencing. Henceforth, I strongly recommend a thorough review and proofreading of the manuscript to correct these typographical errors.
Requested changes
Major concern:
1. Please provide further clarification on their motivation for choosing this particular model with anisotropic hopping parameters.
2. Concerning the phase diagram, the region claimed to exhibit T-linear behavior seems too broad, and the evaluation of the boundary is unclear, particularly around T ~ 0.08. The manuscript states, “There is a slight deviation from the T-linear regime seen in Fig. 2a) for T ~ 0.08. The raw data for the scattering rate at p=0.06 in Fig.3 shows that this deviation from T-linearity is barely noticeable (at page 4, right column)." However, this deviation is not easily discernible, potentially due to the relatively small size of the figure and the fact that it is not plotted on a logarithmic scale, which may obscure deviations from T-linearity. To address this concern and clarify the observations, it would be prudent to include a version of Fig. 3 plotted on a logarithmic scale. Additional it would be helpful how authors determined the boundary of T-linearity in the phase diagram.
Minor concern:
[Contents-Related Concerns]
1. The explanations for \tilde{K} and the six patches K_i are insufficient. An explanation should be added at the end of the first paragraph in Section IIB, where Fig. 1b is mentioned, to enhance readability for the reader.
2. Regarding “For this reason, we limit ourselves to T lower than 0.2” on page 3 right column. It is not very clear since the energy unit is not clearly defined.
3. To facilitate a clearer understanding of the manuscript, I recommend that the hole doping parameters p and x be thoroughly defined and explained prior to Sec. III. This would ensure that readers have a solid grasp of these key concepts before delving into the subsequent analysis and discussion.
[Typesetting and Formatting Concerns]
4a. In the first paragraph of Sec. IIC, "he" should be corrected to "the".
4b. Plese modify a reference error on page 4 in the right column where "Table II C" is mentioned; it should instead refer to "Table I".
The full reply is in the attached file.
Attachment:

---

## Round 1 · Referee Report · Thomas Schäfer (Referee 3) · 2024-1-22

Strengths
- The manuscript presents an in-depth analysis of the scattering rate for one of the paradigmatic models for electronic correlations, the Hubbard model, on a triangular lattice.
- The finding of two different T -linear regimes and their attribution to two different physical mechanisms is very intriguing.
- Cutting-edge quantum embedding methods are used to perform the calculations presented.
- The conclusions of the manuscript represent fair assessments of the data analyzed.
Weaknesses
- The employed dynamical cluster approximation (DCA), albeit respecting translational invariance, bears discontinuities of the self-energy in the Brillouin zone.
- A full cluster-size dependence (and, therefore, the extrapolation to the thermodynamic limit) could not be given (due to its computational demand).
Report
In their manuscript “Two T-linear scattering-rate regimes in the triangular lattice Hubbard model” the authors Jérôme Fournier, Pierre-Olivier Downey, Charles-David Hébert, Maxime Charlebois, André-Marie Tremblay analyze the scattering rates Γ of the paradigmatic model for electronic correlations, the Hubbard model, on a triangular lattice by means of the dynamical cluster approximation. They find two different regimes as a function of hole-doping in which Γ displays a linear-in-T behavior. The authors relate this intriguing dependence to two different mechanisms: (i) Mott physics at low doping values near the co-called “Sordi-transition”, where also ω/T-scaling is found, and (ii) an interaction-driven mechanism, for which such scaling could not be demonstrated.
We find the content of the paper very interesting and important to many people in the field of theoretical condensed matter physics. The topic of non-Fermi liquid behavior and quantum criticality at strong coupling represents a long-standing problem in solid state theory, to which this manuscript presents a significant contribution. As such, the manuscript fulfills the requested criterion of “open a new pathway in an existing or a new research direction, with clear potential for multipronged follow-up work”. The paper is written in a clear language and the results obtained and presented therein seem to be scientifically sound. Further, the manuscript collects a wide variety of criteria for studying the electronic normal-state regimes on the single-particle level, giving methodological value to the scientific community. We therefore will recommend the manuscript for publication in SciPost Physics, after the following comments and change requests have been properly addressed.
Requested changes
Main points 1. While we understand that the cluster size Nc of the calculations is limited due to computational constraints, we were wondering whether there is a principle threshold of Nc where the linear-in-$T$ behavior becomes apparent. Did the authors also try to employ smaller clusters (e.g., $N_c = 3$)? If yes, did the linear-in-$T$ behavior persist for both regimes? 2. Connected to the first point: as far as we understand, the linear-in-T behavior of the resistivity emerging from (quantum) critical properties of the Mott metal-insulator transition can be found in single-site ($N_c = 1$) dynamical mean-field theory, see Ref. 17 of the manuscript. Does this mean, in turn, that only for regime (ii) [i.e., the interaction-driven mechanism] an actual cluster calculation ($N_c > 1$) is necessary? 3. Can the authors comment (for one representative data point) on the spin and charge correlation lengths in the two $T$-linear regimes or is this computationally out of reach? 4. As sort of general orientation it would be interesting to see the Mott transition (including the coexistence regime) mapped out for the doped system and plotted within Fig. 2. Do the authors think that this is numerically feasible? 5. To demonstrate the existence of a pseudogap (as indicated in paragraph 2, left column, of page 5), it would be very instructive to show the spectral weight at the Fermi level [from, e.g., an extrapolation of $A(\mathbf{K},i\omega_n → 0$)] for different $\mathbf{K}$-patches in this regime. 6. In the interest of graphical clarity: did the authors consider log-log plots for the scattering rates, where one could read off also exponents possibly different from 1? 7. Could the authors make a more explicit comment on the mapping to the electron doped side, if possible? From Fig. 4 the mapping seems to be non-trivial.
Minor points (and typos, etc.) 8. To avoid possible confusion with other studies, it would be helpful if the authors could show the (non-interacting) density of states for their model [e.g., as Fig. 1c)]. 9. In order to judge the overall level of scattering and the degree of momentum differentiation, a plot of Γ for the respective momentum patches, at constant $T$ and varying $p$ would be helpful as amendment to Fig. 2. 10. Fig. 2: are the temperature ranges shown in (a) and (b) the same? If yes, this should be stated in the caption. 11. Fig. 3: We congratulate the authors to this very fine-grained high quality data. We have the following requests/questions: (a) As the doping regime from p = 0.06 to p = 0.08 is particularly interesting, why are only the endpoints of this regime shown in Fig. 3? Can the authors confirm that the qualitative behaviour does not change in between these points? (b) In Fig. 4, the notion of the yellow and red stars are introduced as the critical interactions of the metal-to-insulator transition. We think it could be worthwhile to mark them in Fig. 3 at the respective fillings/temperatures. 12. Fig. 4: (a) Which $\mathbf{K}$-patches do the spectral functions correspond to? Are these local ones? This should be stated in the caption and the spectral function y-axis labels should be adapted accordingly. (b) The authors give the spectral function for the $T$-linear ”Mott-driven” regime. What is the splitting of the Hubbard bands? Would it be worthwhile to give the spectral function for a representative point of the ”interaction driven” $T$-linear point for comparison? Do you still find Hubbard bands, does the splitting persist? (c) Further, we consider the colour coding confusing, especially in direct comparison with Fig. 3. Could the authors choose different colors and/or line styles for the fillings in the inset of Fig. 4 to discriminate from different patches in Fig. 3? If the spectral functions correspond to a certain $\mathbf{K}$-patch, it would be worthwhile to use the same colour coding here as in Fig. 3. 13. Fig. 5: We could not find data for p = 0.5. 14. Appendix A: How does the cluster geometry for $N_c = 12$ look like? 15. Fig. 8: $U_{c3}$ is not defined in the current manuscript. 16. Typos (a) p. 1, left column: “on scattering” → “on the scattering”, “value of” → “values of”. (b) p. 3, left column: “One of he” → “One of the”. Right column: “higher the” → “higher than the”. (c) p. 5, right column: “near that T = 0.05” → “near T = 0.05”. (d) p. 8, left column, formatting error: A point ”.” went to a new line instead of remaining at the end of the respective sentence. (e) p. 8, left column: “disorder on scattering” → “disorder on the scattering”.

---

## Round 2 · Referee Report · Thomas Schäfer (Referee 3) · 2024-7-15

Report
The authors answered to all our concerns and suggestions in a very detailed and satisfying manner and adjusted their manuscript accordingly.
We hence recommend the updated version of the manuscript for publication in SciPost Physics.
Recommendation
Publish (surpasses expectations and criteria for this Journal; among top 10%)

---

## Round 2 · Author Response

We are grateful for the thorough and very insightful review of our manuscript. We have carefully considered every comment when making changes to the paper, and we now believe this current version is satisfactory regarding your previous suggestions. A PDF was included for response to each referee.

---

## Round 2 · List of Changes

We specified how hole and electron doping were obtained.
We explained how the results on the different patches $\mathbf{K_i}$ were obtained by averaging the results on the different $\Tilde{\mathbf{k}}_j$ within the patch.
A new subsection was added to describe all the limiting factors associated with our chosen method.
A figure of the quasiparticle scattering rate as a function of temperature at doping $p=0.06$ was added in Appendix B. A comment on the value of $\alpha$ at $p=00.6$ was also added in section III.A.
A figure of scattering rate as a function of temperature at $p=0.25$ and $p=0.17$ for different cluster sizes is added in Appendix A. Those results changed our interpretation of the $T^2$ scattering rate at $p\sim 0.17$, which was found on the first paragraph of section III.B. From these results, a paragraph about the importance of superexchange in the interaction-driven regime was added to section IV.A.
Another version of the phase diagram on Fig. 2a, displaying the Sordi transition and the Widom line at $U=8.4$, is added to Appendix C.
A figure of the scattering rate as a function of temperature for the square lattice ($t'=0$) and the triangular lattice $t'=1$, is added to the discussion. The results from this figure are used to justify our comparison with the square lattice in section IV.B.
A figure of the scattering rate as a function of temperature for $p=0.04$, $p=0.06$ and $p=0.08$ at $U=8.4$, and $p=0.25$ at $U=8.5$, is added to appendix E. This allows to distinguish better the different regimes of the scattering rate.
A figure of the scattering rate as a function of doping at $\beta=20$ and $U=8.4$ is added to appendix E.
A figure of the spectral weight as a function of temperature for $p=0.023$, $p=0.04$ and $p=0.06$ at $U=8.4$, and $p=0.25$ at $U=8.5$, is added to appendix F. This allows to demonstrate the existence of the pseudogap.
The spectral weight at $p=0.25$ and $T=1/40$, $1/30$ and $1/20$, for both $U=8.5$ and $U=6.0$, is added to appendix F. The spectral weights are accompanied by a figure of the quasiparticle weight $Z$ as a function of temperature, for $U=6.0$ and $U=8.5$, at $p=0.25$.

---

## Editorial Decision

published